# Towards High-Resolution Copy-Evident Ceramic Tiles: A Deep Learning Framework for Halftoning and Watermarking

**Jianfeng Lu** [1,†], **Zhiwen Wang** [1,2,†], **Li Li** [1,*], **Ching-Chun Chang** [3] , **Ting Luo** [4] and **Wei Gu** [5]

1 Department of Computer Science, Hangzhou Dianzi University, Hangzhou 310018, China; jflu@hdu.edu.cn (J.L.); wzw2019@hdu.edu.cn (Z.W.)
2 Key Laboratory of Brain Machine Collaborative Intelligence of Zhejiang Province, Hangzhou 310018, China
3 Department of Computer Science, University of Warwick, Coventry CV47AL, UK; ching-chun.chang@warwickgrad.net
4 Department of Academic Research, College of Science and Technology, Ningbo University, Ningbo 315000, China; luoting@nbu.edu.cn
5 Department of Computer Science, Anhui University, Hefei 230039, China; 09055@ahu.edu.cn
* Correspondence: lili2008@hdu.edu.cn; Tel.: +86-1366-663-2945
† These authors contributed equally to this work.

**Abstract:** Ceramic art is essential in interior design and decoration, and making exquisite ceramic tiles imposes strict requirements for inkjet printing technology. High-resolution ceramic tiles are often produced through inkjet printing, in which the input images are converted into a halftone format. However, traditional binary halftoning techniques cannot produce high-resolution images for the ensuing printing process. Given that the processes of inkjet printing and high-temperature firing of ceramic tiles are a highly complex nonlinear system, and existing halftoning methods pose intractable problems, including inconsistent textures and color deviations. Based on a modified U-Net model and a modified error diffusion algorithm, we propose a multilevel halftoning method, which is capable of converting color-separation images of ceramic tiles into high-resolution halftone images. To deter copyright infringement, we further apply an ad hoc invisible watermarking method for halftone images. In this paper, we propose a four-stage framework: (1) A self-built dataset is used to solve non-convergence and overfitting problems caused by the unbalanced samples and non-uniqueness of halftone images. (2) A modified U-Net model is trained on the self-built dataset and applied to the ceramic-tile images. (3) An improved error diffusion algorithm is used to calibrate and convert the predicted continuous-tone transition images into multilevel halftone images for inkjet printing. (4) A invisible and robust watermark is embedding algorithm towards halftone images is proposed for copyright protection. Experimental results show that our methodology is effective for performing the color-to-halftone transformation and identifying the copyright.

**Keywords:** error diffusion; image stitching; invisible watermarking; multilevel halftoning; U-Net

## 1. Introduction

In the whole process of printing, ensuring the fidelity of color after the transmission of electronic images between devices is a complex problem in the industry. Most digital output devices are monochrome devices, such as inkjet printers and digital printers, whereas original electronic images are continuous-tone images. Most of these binary devices adopt halftone technology to enable the reproduction of authentic electronic images. Halftone refers to quantifying a continuous-tone image into a binary image or only a few colors of a color image. Quantized and original images have a similar visual effect. Some well-known halftone algorithms include the ordered dither [1,2], error diffusion [3], dot diffusion [4–6], iterative processing [7], and Lut [8].

The improvement of the visual requirements of printed images has shown that the halftone image quality can be enhanced by improving the tonal performance of individual

pixels. An intermediate tone between the traditional binary (black and white) can make the transition between halftone image more natural, and its quality is improved. Thus, several multilevel halftoning methods [9–15] have emerged. In 1965, Huang et al. put forward a multilevel dithering halftone algorithm [9]: adding a small amount of intermediate halftone in binary halftone images can remarkably improve the quality of halftone images. Experiments show that even adding one or two intermediate halftones can substantially enhance the image's overall visual perception. In 1997, Spaulding et al. implemented the multilevel halftoning method on the basis of the ordered dithering algorithm [10,11]. Wong et al. proposed an embedded multilevel error diffusion algorithm [12]. The idea is embedding a binary halftone into a multilevel halftone. To some extent, this algorithm can eliminate artificial traces, but the quality of the whole halftone image decreases, thereby weakening the advantage of the multilevel error diffusion. In 2008, Rodríguez Arce et al. proposed a multilevel blue noise model for multilevel halftones [13]. Suetake et al. proposed a relatively simple multilevel halftone algorithm on the basis of the error diffusion to solve banding artifacts [14]. The two algorithms decompose the input image into layers, use a binary error diffusion technique to process each layer halftone, and finally, combine each layer's halftone processing results to generate a multilevel halftone image. In 2010, Fung et al. proposed a multilevel halftone using the multiscale error diffusion [15]. The algorithm's output has good blue noise characteristics, no directional artifact, and retains the original image's feature details.

Given the wide use of the digital halftone technology and the development of the neural network, K.R.Crounse et al. proposed a halftone algorithm on the basis of the convolutional neural network (CNN) in 1993 [16]. Compared with that obtained using the error diffusion algorithm, the halftone image generated by the halftone algorithm more realistically reproduces the primary color. In 1997, P.R.Bakic et al. proposed a multilevel halftone algorithm for digital images on the basis of the CNN [17], and the results of this algorithm were verified in human subjective quality experiments. In 2008, Huang et al. proposed a hybrid algorithm of digital halftone and inverse halftone on the basis of the neural network [18]. J.Guo et al. proposed a generative adversarial network model [19], which can be used in halftone and structural reconstruction.

In the application scene of ceramic tile inkjet printing, the quantification process cannot be described as an ideal mathematical model because the processes of inkjet printing and high-temperature firing of ceramic tile are a highly complex system. Moreover, converting a color separation image into a halftone image by using existing binary and multilevel halftone technology is impossible because of the high precision requirement of inkjet printing a ceramic tile. When using Yik-hing Fung's error diffusion method [15] to process and transform the ceramic tile color separation image into a multilevel halftone image, we could obtain an example result, as shown in Figure 1b. Comparing Figure 1b with the original image (Figure 1a), we found they have a similar texture but a remarkable deviation in color. As traditional methods could not obtain satisfying result, we try to introduce deep neural networks for the multilevel halftoning task. In this paper, we employ the classical U-Net framework [20–22], which is quite suitable to be applied on a small sample, and fine tune it for quantifying a 0–255 tile color separation image into a 0–1 transition image. The result of quantization into multilevel halftone using U-Net is shown in Figure 1c. As can be seen from the result, only a portion of the features are learned, which results in large gap to the original image both in texture and color. This paper proposes a modified U-Net model and error diffusion framework to implement the conversion of color separation images into multilevel halftone images, which are suitable for ceramic tile inkjet printing. The rendered result after calibrating the modified U-Net model prediction and error diffusion algorithm is given in Figure 1d, which is nearly the same with the original image.

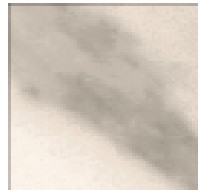 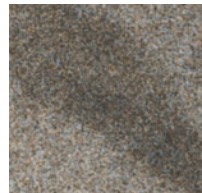 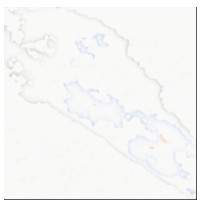 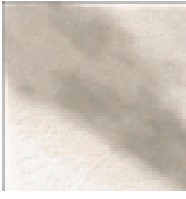

(**a**) Original image    (**b**) Yik's method    (**c**) U-Net method    (**d**) Our method

**Figure 1.** Rendering results of different multilevel halftone methods.

Digital watermarking [23,24] is a technology to embed specific information into digital products to protect the copyright or integrity of digital products. In order to protect digital products and prevent tile halftone image from being illegally copied by others, this paper embeds a digital and invisible watermark into a ceramic tile multilevel halftone image. The existing methods [25,26] can embed visible watermarks in binary halftone images. However, these methods are not used in tile printing scenes. Because of the quality problem of tile printing, we need an invisible watermark-embedding method suitable for multilevel halftone. Therefore, we propose a watermark-embedding method for tile scenes.

The main contributions of this paper are as follows.

(1) A dataset is built. Given that the data of the ceramic tile halftone image have the characteristics of uneven sample classification and nonunique halftone results, the problems of nonconvergence and overfitting of model training easily occur when the ceramic tile data are trained directly. Therefore, some images in Corel5K are collected to form a dataset in this paper. The dataset is prepared after preprocessing to deal with the problems of model training nonconvergence and overfitting.

(2) The loss function, parameters, and structure of the original U-Net are modified to predict the color separation image of the ceramic tile to the halftone image. The size of the input, number of output channels, activation function of the last layer, loss function, and filling method of convolution are modified to make U-Net suitable for the ceramic tile application scene.

(3) Network output data are calibrated on the basis of image stitching and the error diffusion algorithm. Image stitching solves the block effect when small pieces of a continuous-tone transition image are integrated into a complete continuous-tone transition image. The modified error diffusion algorithm is used to transform the continuous-tone transition image into a multilevel halftone image suitable for ceramic tile printing.

(4) A method to prevent tile the halftone image from being copied by watermark embedding is proposed.

The remaining part of this paper is organized as follows. Section 2 introduces the self-built dataset, preprocessing of data, and verification of channel independence. Section 3 introduces our method, including the prediction part's network, the calibration part's image stitching and error diffusion algorithm, and the preventing copy part's watermark-embedding algorithm. Section 4 shows the experimental results and related parameters. Section 5 provides the conclusion.

## 2. Data Preparation

The high-fidelity printing of ceramic tile design drawings is composed of two steps. A ceramic tile image is converted into a ceramic tile color separation image by using the International Color Consortium (ICC) Profile, and a ceramic tile color separation image is adjusted to a halftone image suitable for a ceramic tile inkjet printer. The first step (i.e., ICC Profile) is provided by the inkjet printer manufacturer. The user can easily convert the design drawing to the color separation image through the commonly used image processing software. For the second step, different manufacturers have their algorithms privately. For the same tile color separation image, different manufacturers obtain a variety

of halftones. Therefore, a general algorithm should be studied to realize the conversion of color separation images into halftone images.

Considering the successful application of machine learning algorithms in the image-to-image translation task, this paper intends to use machine learning methods to realize the above conversion. The construction of the dataset is the first step. Given the nonuniform sample classification and the nonunique halftone result of the halftone image of the ceramic tile, a dataset with similar simple texture characteristics, rich color, and relatively balanced category is constructed. The data set prevent non-convergence and over-fitting problems in the training model. The data enhancement method based on the fuzzy operation is used to alleviate the problem caused by the halftone results' nonunique characteristics. Simultaneously, the channel independence between tile color separation image and halftone image are fully used to train the channel model separately.

### 2.1. Dataset

For a fixed ceramic tile production line, color separation image and corresponding halftone images are collected in pairs. The color separation image of ceramic tile is quantized into color separation image, and its quantized value is 0, 1, 2 and 3. Figure 2a is a Ceramic tile color separation image, and Figure 2b is a data distribution image of Figure 2a. Figure 2c is a landscape image, and Figure 2d is a data distribution image of Figure 2c. After analyzing a lot of sample images, we found these tile images have very simple tone and texture, corresponding halftone images are largely consisted of 0 and 1 data. Thus, the classification of the halftone image data of the tile are not uniform. Comparing Figure 2b with Figure 2d, we found the distribution of landscape image is more uniform than the distribution of ceramic tile image data. Therefore, this paper constructs the dataset to solve the data distribution not uniform problem.

A total of 3000 landscape images are selected from the Corel5K dataset in this paper. This dataset covers images that have rich colors for training a neural network model. Moreover, the selected images have simple backgrounds and share similar characteristics with ceramic tile images to some extent. All images are resampled to 256 × 256 pixels, and the warm and cold tones are adjusted for sample expansion to obtain 9000 images. According to the ICC profile standard, these images are converted into color separation images, and corresponding halftone images are used to form the training set. The collected tile images are divided into 256 × 256 pixel images as the test set.

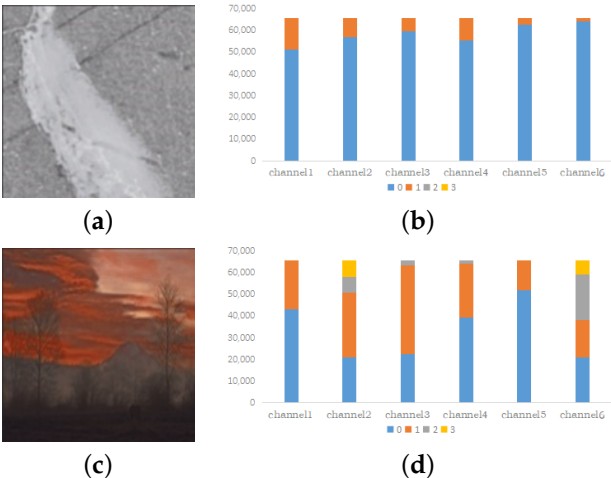

**Figure 2.** The distribution of halftone data for each channel of tile and landscape images. (**a**) Ceramic tile color separation image; (**b**) Ceramic tile color separation image's data distribution image; (**c**) Landscape; (**d**) Landscape color separation image's data distribution image.

## 2.2. Data Preprocessing

The same continuous-tone value of the color separation image corresponds to the different values of the halftone image, directly using the halftone image for training results in the non-uniqueness of input samples and labels during model training. In this paper, the mean filter fuzzy processing technology is used to enhance the training set. Specifically, the $5 \times 5$ mean fuzzy filter is as follows:

$$K = \frac{1}{25} \times \begin{bmatrix} 1 & 1 & 1 & 1 & 1 \\ 1 & 1 & 1 & 1 & 1 \\ 1 & 1 & 1 & 1 & 1 \\ 1 & 1 & 1 & 1 & 1 \\ 1 & 1 & 1 & 1 & 1 \end{bmatrix} \tag{1}$$

## 2.3. Channel Independence Analysis

The traditional four-color printing uses yellow, magenta, blue, and black inks to copy the original printing color. The ceramic tile uses spot color printing, and the spot color ink is premixed using the ceramic tile factory or ink factory production. In traditional printing, the information of yellow, magenta, cyan, and black channels is used to control the ink output of the corresponding nozzle, and the channels are independent of each other. Experiments show that the channel of the spot color printing of ceramic tile has independent characteristics. The channels of the color separation image and halftone image have a one-to-one correspondence and do not interfere with each other. Figure 3a is a gradual color separation image. From top to bottom, each color takes up one channel. From left to right, image's value increases from small to large. Figure 3b–g correspond to the halftone image of each channel from 1 to 6. By comparing Figure 3b–g with Figure 3a, we found that each color of the color separation image occupies one channel, and the corresponding halftone image also occupies one channel for each color. The experimental results prove a one-to-one correspondence between the channels of the color separation image and the channel of the halftone image.

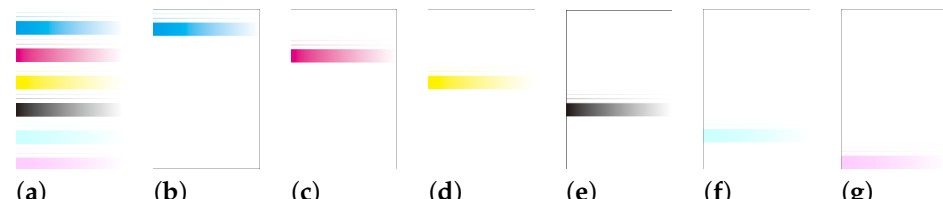

(a)      (b)      (c)      (d)      (e)      (f)      (g)

**Figure 3.** Experimental results of channel independence. (**a**) Gradual color separation image; (**b**) First; (**c**) Second; (**d**) Third; (**e**) Fourth; (**f**) Fifth; (**g**) Sixth.

## 3. Proposed Method

The modified U-Net and error diffusion method are proposed to implement the conversion of ceramic tile color separation images into inkjet-printing halftone images. The general flowchart of this method is shown in Figure 4. First, the U-Net is trained on the landscape dataset. Then, the trained model is migrated to the tile dataset for the block prediction of the tile halftone image, and the small block set of continuous-tone transition images is obtained. Next, the small block set of continuous-tone transition images is stitched to a complete continuous-tone transition image. Furthermore, the modified error diffusion algorithm is used to generate a multilevel halftone image. Finally, the multilevel halftone image is embedded with watermark.

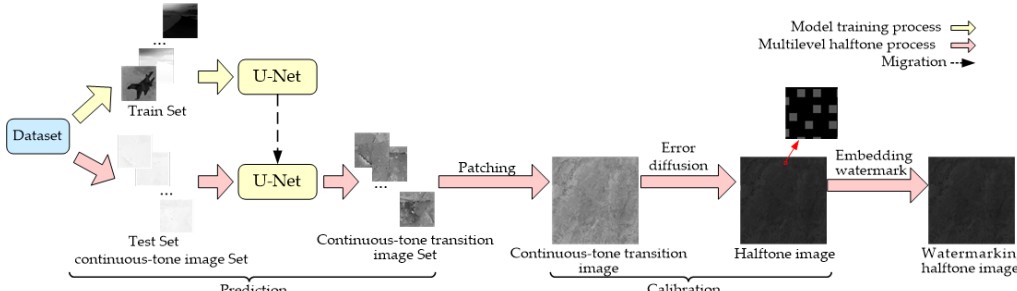

**Figure 4.** General flowchart.

### 3.1. Prediction Phase

The U-Net [20] is convolutional network architecture for fast and precise segmentation of images and is widely used in machine learning. It has the advantages of strong adaptability and small sample training. Considering that the original U-Net's input and output sizes are fixed, the tile image should be divided into small pieces of images with a uniform size, and these pieces are stitched back to the original size after the U-Net model's prediction. Considering that the channels of the tile color separation image and halftone image are independent of each other, each channel is trained separately to obtain each channel's model.

The original U-Net model consists of contracting (left) and expansive (right) paths. The contracting path consists of two $3 \times 3$ unpadded convolutions. Each convolution is followed by a rectified linear unit (ReLU), and a max-pooling of (2,2) operation with stride 2 is performed for downsampling. In each downsampling, the number of feature channels is doubled. Each step in the expansive path includes the upsampling of feature maps followed by the $2 \times 2$ upconvolution, which halves the number of feature channels and connects them with the corresponding cropped feature map in the contracting path. Two $3 \times 3$ convolutions are performed, each of which is followed by a ReLU. At the last layer, each 64-component feature is mapped to the desired number of classes by using the $1 \times 1$ convolution.

In this paper, the U-Net is modified, and the color separation image of 0–255 is predicted to be a continuous-tone transition image of 0–1. The network structure is listed in Figure 5. In the modified network, the input size is changed to $256 \times 256$ pixels. The network output is changed from two channels to one channel. The last softmax of the original network is replaced with a sigmoid. The unfilled convolution (unpadded convolutions) of the original network is replaced with a filled convolution (padded convolution) to produce images with the same size. The loss function of the network is changed from cross-entropy loss to mean-square-error (MSE), and the formula is:

$$MSE = \frac{1}{n} \sum_{i=1}^{n} (y_i - \hat{y}_i)^2 \qquad (2)$$

$y_i$ is the halftone image after mean blur processing, $\hat{y}_i$ is the predicted continuous-tone transition image, and $i$ represents the i-th sample.

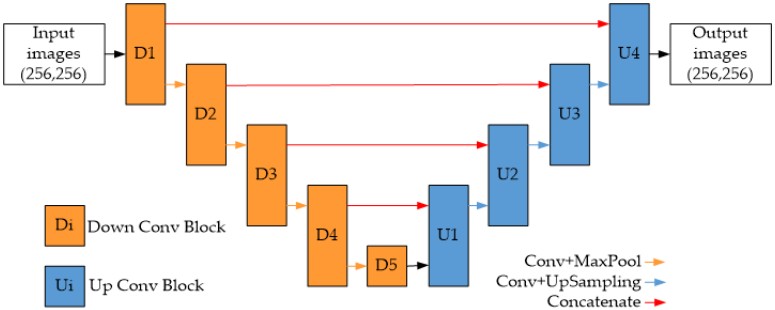

**Figure 5.** The U-Net structure.

The parameters of the network are shown in Table 1.

**Table 1.** The U-Net composition of the network.

| Block | Layer (Type) | Output Shape | Param |
|---|---|---|---|
| | InputLayer | (256,256,1) | 0 |
| D1 | Conv2D+relu | (256,256,64) | 640 |
| | Conv2D+relu | (256,256,64) | 36,928 |
| | Maxpooling2D | (128,128,64) | 0 |
| D2 | Conv2D+relu | (128,128,128) | 73,856 |
| | Conv2D+relu | (128,128,128) | 147,584 |
| | Maxpooling2D | (64,64,128) | 0 |
| D3 | Conv2D+relu | (64,64,256) | 295,168 |
| | Conv2D+relu | (64,64,256) | 590,080 |
| | Maxpooling2D | (32,32,256) | 0 |
| D4 | Conv2D+relu | (32,32,512) | 1,180,160 |
| | Conv2D+relu | (32,32,512) | 2,359,808 |
| | Dropout | (32,32,512) | 0 |
| | Maxpooling2D | (16,16,512) | 0 |
| D5 | Conv2D | (16,16,1024) | 4,719,616 |
| | Conv2D | (16,16,1024) | 9,438,208 |
| | Dropout | (16,16,1024) | 0 |
| U1 | UpSampling2D | (32,32,1024) | 0 |
| | Conv2D+relu | (32,32,512) | 2,097,664 |
| | Concatenate | (32,32,1024) | 0 |
| | Conv2D+relu | (32,32,512) | 4,719,704 |
| | Conv2D+relu | (32,32,512) | 2,359,808 |
| U2 | UpSampling2D | (64,64,512) | 0 |
| | Conv2D+relu | (64,64,256) | 524,544 |
| | Concatenate | (64,64,512) | 0 |
| | Conv2D+relu | (64,64,256) | 1,179,904 |
| | Conv2D+relu | (64,64,256) | 590,080 |
| U3 | UpSampling2D | (128,128,256) | 0 |
| | Conv2D+relu | (128,128,128) | 131,200 |
| | Concatenate | (128,128,256) | 0 |
| | Conv2D+relu | (128,128,128) | 295,040 |
| | Conv2D+relu | (128,128,128) | 147,584 |
| U4 | UpSampling2D | (256,256,128) | 0 |
| | Conv2D+relu | (256,256,64) | 32,832 |
| | Concatenate | (256,256,128) | 0 |
| | Conv2D+relu | (256,256,64) | 73,792 |
| | Conv2D+relu | (256,256,64) | 36,928 |
| | Conv2D+sigmoid | (256,256,1) | 65 |

### 3.2. Calibration Phase

The calibration phase follows a two-step pipeline: a patchwork treatment that alleviates the blocking artifacts when stitching small patches together and an error diffusion that transforms a continuous-tone image into a multilevel halftone image.

### 3.2.1. Image Stitching

If the small continuous-tone transition image is stitched directly, evident cracks appear. The following methods are adopted to eliminate cracks.

Step 1: The new boundary pixel value is expanded by the edge pixel value as shown in Figure 6. Figure 6a, Figure 6b and Figure 6c represent the original color separation image, the upper boundary extended color separation image and the final boundary

extended color separation image, respectively. H represents the height of the image, W represents the width of the original image, and X represents the pixel width of the extended boundary.

Step 2: The final boundary extended color separation image is segmented to obtain the small block set of the color separation image, in which the segmenting is performed in the way of first rows and then columns, and the step size is (256−X).

Step 3: The color separation image block set obtained in Step 2 is used to obtain the continuous-tone transition image block set through our model.

Step 4: The middle (256−2X) × (256−2X) pixels of each predicted continuous-tone transition image are matted and stitched to form a complete continuous-tone transition image step by using first rows and then columns.

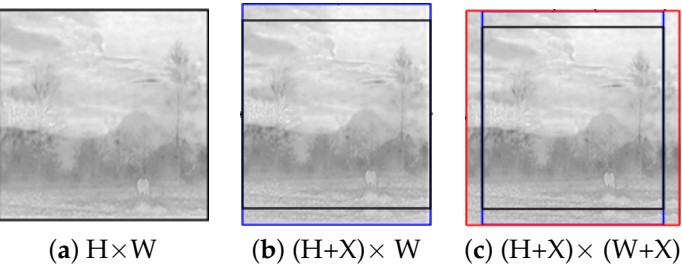

**(a)** H×W      **(b)** (H+X)× W      **(c)** (H+X)× (W+X)

**Figure 6.** The extent of color separation image.

### 3.2.2. Error Diffusion

Floyd and Steinberg proposed an error diffusion algorithm [3] in 1975. The error diffusion algorithm starts from the first pixel in the continuous-tone image, quantifies the current pixel value to 0 or 1, and allocates the quantization error to the adjacent pixel certain proportion. This process is repeated for the subsequent pixels in the order of scan lines until the last pixel is processed. Floyd and Steinberg used a filter to assign quantization errors (Figure 7). Currently quantized and processed pixel points are denoted by "*" and "-", respectively, and the values in adjacent positions represent the allocation ratio of quantization errors.

| - | * | 7/16 |
|---|---|---|
| 3/16 | 5/16 | 1/16 |

**Figure 7.** Floyd and Steinberg error diffusion filters.

The quantization formula of pixel point $P(i, j)$ is defined as follows:

$$P(i,j) = \frac{7}{16}P(i,j+1) + \frac{3}{16}P(i+1,j-1) + \frac{5}{16}P(i+1,j) + \frac{1}{16}P(i+1,j+1) \qquad (3)$$

The continuous-tone image can only obtain 0 and 1 color levels through the traditional error diffusion algorithm. However, the ceramic tile inkjet printing requires a halftone image of 4 color levels (0, 1, 2, 3, respectively). Therefore, based on the error diffusion algorithm, the continuous-tone image is transformed into a multilevel halftone image. The modified error diffusion algorithm is described as follows.

Input: predicted continuous-tone transition image

Step 1: Find the maximum value of the continuous-tone transition image and denote it as max.

Step 2: Normalize the continuous-tone transition image to [0, 1] by maximum and minimum.

Step 3: Multiply the (0, 1)-normalized continuous-tone transition image by max.

Step 4: Allocate quantization errors with Floyd and Steinberg filters to obtain error diffusion.

Step 5:   If error diffusion is greater than 3, truncate the error diffusion to 3.
Output:multilevel halftone image

### 3.3. Watermark Embedding

After obtaining the multilevel halftone image by using the error diffusion algorithm, a watermark is embedded to prevent the halftone image from being copied. The main embedding process is listed as follows:

Input:   pmultilevel halftone image with the size of 256×256, 16-bit watermark $w_i$ ($0 < = i <= 15$ to be embedded).
Step 1:   Divide the multilevel halftone image into 16 blocks with the size of 64×64.
Step 2:   Embed $w_i$ in the first element of the i-th Block in turn. If the embedded bit is 0, the first element of the Block is changed to 0; otherwise, it is changed to 1.
Output:Watermarked halftone image.

## 4. Results and Analysis

To verify the effectiveness of the proposed method, the following experiments are designed: comparision with other methods, experiment of image stitching, assessment of inkjet consumption, evaluation of the quality of generated images, evaluation of actual printing results, and experiment of embedding invisible watermark.

### 4.1. Experimental Setup

The network model is trained and tested on the NVIDIA 2080 Ti. In the model training, the random optimization algorithm Adam is used. The learning rate, same number of iterations for each channel, and batch size are 1E–4, 20, and 20, respectively. The average single iteration time is about 234 s, and the model training for each channel takes about 80 min. In the image stitching, the boundary expansion pixel parameter X is set to 48.

### 4.2. Comparison with Other Methods

There are several articles for halftone quantization. Yik's method [15] is applicable to multilevel halftone images. Guo's method [19] is applicable to binary halftone images. There are also several articles that could embed watermarks into binary halftone images. Sun et al. [25,26] are all based on binary halftone visible watermarking. However, there is no research that could simultaneously solve the problems of generating multilevel halftone image and embedding an invisible watermark into a multilevel halftone image is shown in Table 2.

**Table 2.** A list of the relevant halftone methods.

| | Multilevel Halftone | Binary Halftone | Visible Watermark | Invisible Watermark |
|---|:---:|:---:|:---:|:---:|
| Yik's [15] method | ✓ | | | |
| Guo's [19] method | | ✓ | | |
| Sun's [25] method | | ✓ | ✓ | |
| Guo's [26] method | | ✓ | ✓ | |
| Our method | ✓ | ✓ | | ✓ |

As multilevel halftoning is our main focus, we compare our method with Yik's [15] method. Six ceramic tile samples are randomly selected from the experimental results, and the effect of rendering is shown in Figure 8. Through Yik's method [15], U-Net is quantified as 0–1 and then magnified to 0–3, and the modified error diffusion method is compared using the original image. It is found that only using U-Net to generate halftone image can only simulate certain texture but cannot learn features for most images. For Yik's method and Floyd's [3] method to generate halftone image, the texture is similar, but

the color deviation is extremely different. Our method to generate a halftone image is very similar to the original image in texture and color and more feasible compared with other methods.

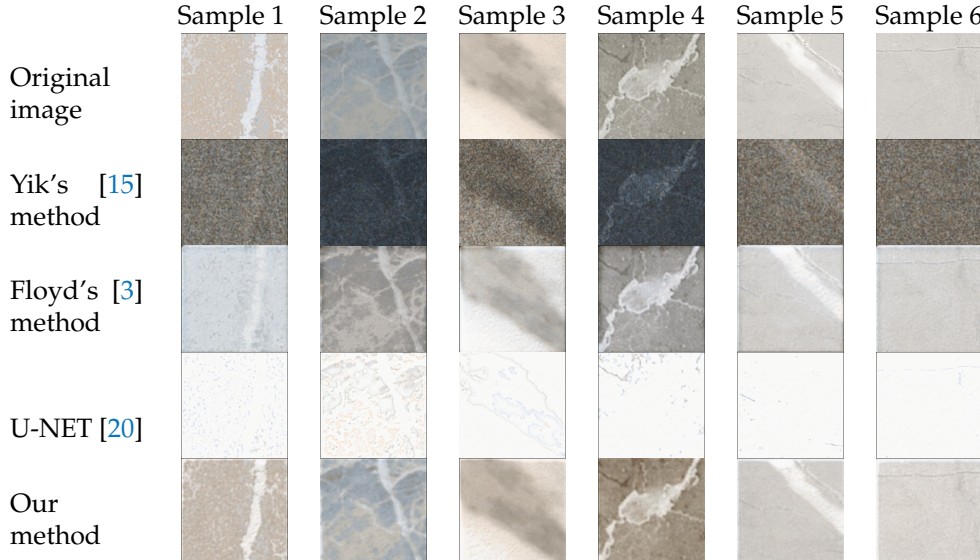

**Figure 8.** A comparison of the different methods.

### 4.3. Experiment of Image Stitching

As the sizes of the produced tiles are inconsistent, we divide the tile images into fixed sized blocks. To save computation resources for training U-Net framework, we design the image block size as $256 \times 256$. As a result, the output tile image pieces from U-Net need to be stitched. However, directly stitching them together could cause gaps between the edge of image pieces. To obtain a better rendering result, we conduct an experiment using our stitching algorithm.

The halftone results without stitching and original halftone printing through ceramic tile inkjet are shown in Figure 9. Evident gaps and block effects after printing the halftone image without stitching are observed.

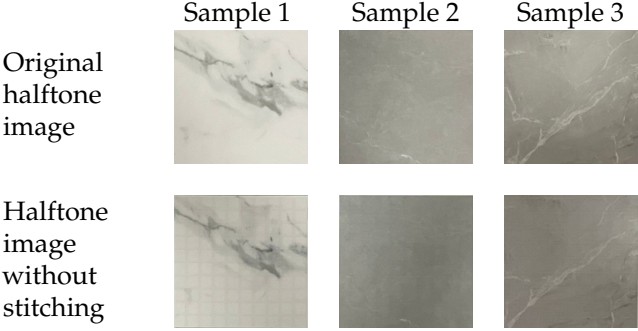

**Figure 9.** A comparison of halftone image results without stitching.

After stitching, the printing result of the ceramic tile is evidently improved. Different extended boundary values are selected to further determine the influence of an extended boundary X on the actual effect, and the experimental effect is shown in Figure 10. The best effect is obtained when the extension boundary is X = 48 pixels.

| Unstitched image | Extended boundary 18-pixel image | Extended boundary 28-pixel image | Extended boundary 48-pixel image |

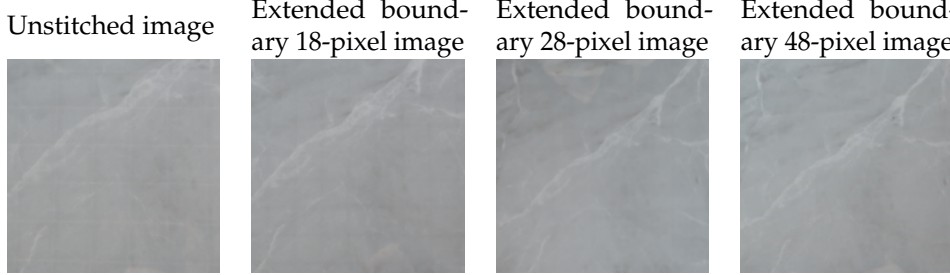

**Figure 10.** A comparison of the boundary value size of sititching.

### 4.4. Assessment of Inkjet Consumption

The ceramic tile inkjet printing controls the amount of inkjet through the halftone image, and the amount of inkjet affects the cost of ceramic tile printing. A high amount of inkjet results in a high cost of ceramic tile printing. Therefore, the scheme's feasibility in this paper is verified by comparing the amount of inkjet of the generated to the original halftone images.

A halftone image is described by the matrix coefficient. The F-norm of the matrix can be used to describe the magnitude of the matrix energy. The F-norm, the arithmetic square root of the square sum of every element in the matrix, measures the distance between a matrix and a zero matrix. The F-norm is defined as follows:

$$\|A\|_F = \sqrt{\sum_{i=1}^{m} \sum_{j=1}^{n} a_{i,j}^2} \tag{4}$$

where $a_{i,j}$ is the value corresponding to the $(i, j)$ coordinates in matrix A.

A high value of F-norm results in high ink consumption and a high printing cost. Therefore, the matrix's F-norm is used to determine the amount of inkjet between the generated halftone image and original halftone image. A comparison of the results is shown in Table 3. THe results show that the sum of the F-norm of the four channels generated by our method is less than the sum of the F-norm of the original halftone image. Thus, a minimal amount of inkjet usage and a decreased cost are observed.

**Table 3.** A comparison of F-norms between the generated halftone images and the original halftone images ($) \times 10^3$.

|  |  | Sample 1 | Sample 2 | Sample 3 | Sample 4 | Sample 5 | Sample 6 |
|---|---|---|---|---|---|---|---|
| Channel1 | Original | 6.96 | 5.14 | 8.48 | 1.15 | 8.95 | 4.79 |
|  | Generate | 7.03 | 5.09 | 7.62 | 1.40 | 8.35 | 4.99 |
| Channel2 | Original | 5.39 | 2.51 | 6.36 | 1.02 | 7.19 | 4.13 |
|  | Generate | 4.44 | 1.61 | 5.81 | 0.53 | 6.52 | 3.02 |
| Channel3 | Original | 4.18 | 4.53 | 2.51 | 3.83 | 6.05 | 4.34 |
|  | Generate | 4.24 | 4.56 | 2.67 | 2.38 | 5.59 | 4.34 |
| Channel4 | Original | 5.81 | 5.09 | 6.50 | 0.64 | 3.56 | 4.52 |
|  | Generate | 5.86 | 5.35 | 5.51 | 1.64 | 3.58 | 4.77 |
| Total sum | Original | 22.35 | 17.27 | 23.85 | 66.45 | 25.74 | 17.78 |
|  | Generate | 21.58 | 16.62 | 21.61 | 5.94 | 24.04 | 17.12 |

### 4.5. Evaluation of the Quality of Generated Images

The Structural SIMilarity (SSIM) is determined to measure the similarity between the halftone images generated by our method and the original halftone images.

$$l(X, Y) = \frac{2\mu_X \mu_Y + C_1}{\mu_X^2 \mu_Y^2 + C_1} \tag{5}$$

$$c(X,Y) = \frac{2\sigma_X\sigma_Y + C_1}{\sigma_X^2\sigma_Y^2 + C_2} \tag{6}$$

$$s(X,Y) = \frac{\sigma_{XY} + C_3}{\sigma_X\sigma_Y + C_3} \tag{7}$$

$l$, $c$, and $s$ are image similarities from brightness, contrast, and structure, respectively. $\mu_X$ and $\mu_Y$ represent the mean $X$ and $Y$ values, respectively, of the image; $\sigma_X$ and $\sigma_Y$ represent the $X$ and $Y$ variances, respectively, of the image; and $\sigma_{XY}$ represents the $X$ and $Y$ covariances of the image. $C1$, $C2$, and $C3$ are constants. $C_1 = (K_1 \times L)^2$, $C_2 = (K_2 \times L)^2$, and $C_3 = \frac{C_2}{2}$ are commonly used to avoid the case that the denominator is 0. In this paper, $K_1$ is 0.01, and $K_2$ is 0.03.

The SSIM comparison results of generated and original halftone images are shown in Table 4. From the table, the SSIM values of each channel of all images are close to 1, and the image is undistorted.

**Table 4.** SSIM comparison results of generating halftone and original halftone.

|          | Sample 1 | Sample 2 | Sample 3 | Sample 4 | Sample 5 | Sample 6 |
|----------|----------|----------|----------|----------|----------|----------|
| channel1 | 0.9939 | 0.9962 | 0.9918 | **0.9997** | 0.9918 | 0.9965 |
| channel2 | 0.9962 | 0.9993 | 0.9947 | **0.9999** | 0.9937 | 0.9978 |
| channel3 | 0.9973 | 0.9969 | 0.9989 | **0.9980** | 0.9952 | 0.9972 |
| channel4 | 0.9953 | 0.9961 | 0.9944 | **0.9997** | 0.9980 | 0.9968 |

### 4.6. Evaluation of the Actual Printing Results

After obtaining the multilevel halftone image by our method and printing it onto the ceramic tile, we evaluate the quality of the printed ceramic tile by observing differences from the original image. The tile printing results of the original and generated halftone images are shown in Figure 11. The picture is slightly different because of the angle and luminosity of the shooting. Our method's halftone image printing result is consistent in texture and slightly biased in color.

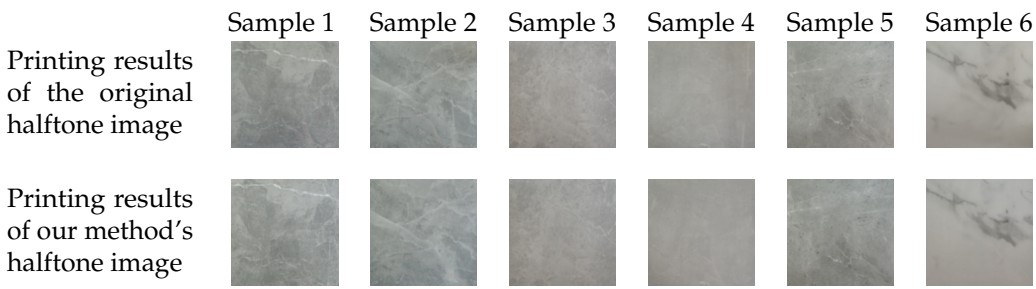

**Figure 11.** Comparison of the results of the original halftone image and the halftone image generated by our method.

### 4.7. Experiment of Embedding Invisible Watermark

The rendering results of the original halftone image and the watermarked halftone image are shown as in Figure 12. In general, the texture and color of the rendering result are basically the same, and the 16-bit watermark information is embedded in it, so as to achieve the purpose of preventing the copy of the tile halftone image.

| | Sample 1 | Sample 2 | Sample 3 | Sample 4 | Sample 5 | Sample 6 |
|---|---|---|---|---|---|---|
| Rendering results of original halftone | | | | | | |
| Rendering results of halftone image embedded watermark | | | | | | |

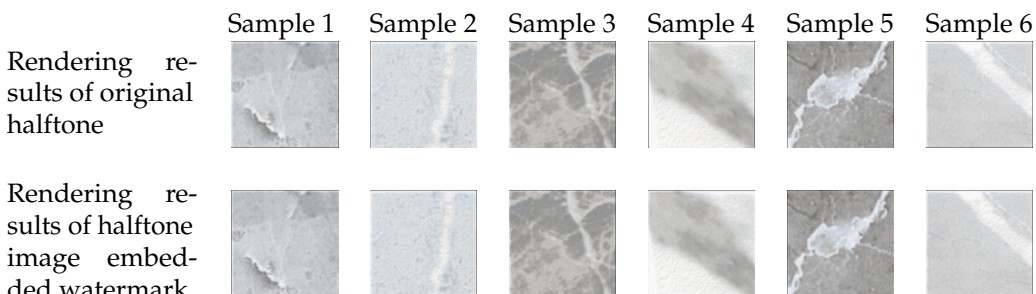

**Figure 12.** Comparison of the results of the original halftone image and the halftone image generated by our method.

### 4.8. Summary Of Experiments

The experimental results show that our multilevel halftone algorithm can convert the ceramic tile's color separation image into a multilevel halftone image and reduce the inkjet amount of ceramic tile while ensuring the authenticity of texture and color. Our watermark algorithm realizes the copyright protection while ensuring the quality of ceramic tiles. Therefore, our algorithm can effectively achieve color to halftone conversion and copyright recognition.

### 5. Conclusions

This paper proposes a framework combining the modified U-Net and the error diffusion algorithm to implement the conversion of a color separation image into the tile inkjet-printing multilevel halftone image. Given the tile halftone image's characteristic with uneven sample classification and nonunique halftone results, the self-built dataset is based on the Corel5K dataset, and the corresponding data preprocessing is carried out. The trained model is migrated to the tile image to implement the prediction from the color separation image to the continuous-tone transition image. Next, the predicted continuous-one transition image is calibrated using the improved error diffusion algorithm. Finally, the multilevel halftone image suitable for ceramic tile inkjet printing is obtained. Experimental results show that this method can convert a color separation image into a halftone image better than existing methods. In addition, compared with existing methods, the amount of inkjet is reduced to a certain extent, and the cost is saved. Furthermore, the proposed invisible watermarking method can prevent the copy of tile halftone image efficiently. In the future, we will try to improve our method through the following aspects: try more deep networks, for example, U-Net ++, to better implement the conversion of ceramic tile color separation image into the tile inkjet-printing multilevel halftone image; inversely simulate the process of converting a multilevel halftone image into a color seperation image to save the design and production time of ceramic tiles.

**Author Contributions:** Supervision, J.L.; writing—original draft preparation, Z.W.; methodology, L.L.; writing—review and editing, C.-C.C., T.L. and W.G. All authors have read and agreed to the published version of the manuscript.

**Funding:** This work was partially supported by the National Natural Science Foundation of China (No.61971247), Public Welfare Technology Research Project Of Zhejiang Province (No.LGF21F020014), Public Welfare Technology and Industry Project of Zhejiang Provincial Science Technology Department (No.LGG19F0-2016).

**Institutional Review Board Statement:** Not applicable.

**Informed Consent Statement:** Not applicable.

**Data Availability Statement:** The training set data used to support the research results can be obtained online, and the test set data is related to enterprises.

**Acknowledgments:** This work was partially supported by the National Natural Science Foundation of China (No.61971247), Public Welfare Technology Research Project Of Zhejiang Province (No.LGF21F020014), Public Welfare Technology and Industry Project of Zhejiang Provincial Science Technology Department (No.LGG19F0-2016).

**Conflicts of Interest:** Authors declare no conflict of interest.

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
