# Peer review of "Towards High-Resolution Copy-Evident Ceramic Tiles: A Deep Learning Framework for Halftoning and Watermarking"

_electronics, doi:10.3390/electronics10151833_

Round 1

Reviewer 1 Report

The authors proposed a multilevel halftoning method by modifying U-Net and an error diffusion algorithm to protect the copyright of halftone images. This manuscript was designed well and written clearly, the study is novel and interesting. However, there are some aspects that have not been sufficiently explained or justified in this manuscript, that should be reviewed, as specified below.

- The references used in this manuscript are mostly old. It is recommended to update it, specifically the introduction section, with data from the last five years as much as possible.

- In figure 2, what are the values between 0 and 3 that indicate the different color bars?

- Could you compare the experimental results of the U-Net method without error diffusion and with error diffusion? (i.e. as shown in Figure 8)

- If possible, could you add more related studies, not only Yik's method (2010)? As mentioned above, the methods presented in the last 5 years are highly recommended.

Reviewer 2 Report

The U-Net term is present in the text and title, but it is not defined before.

It is necessary to reduce the abstract size and not use acronyms in this part of the text. A portion of the abstract can be a move to the introduction section.

Separate the related work from the introduction section to a specific section for this focus is relevant. The section can conduct a textual comparison between your proposal and state of the art in the related work.

The U-Net model is only introduced in section 3.1

A more recent reference than [15] needs to be present to justify the actuality of the research. 

It is necessary to have an end paragraph in section 4.

Where is it possible to access the dataset?

Reviewer 3 Report

Authors suggest new methods use decomposition of image to halftones via auto-encoder model by combining with use error diffusion algorithm.

A very interesting and novel approach.

Few notes I would ask to modify/answer:

* Poor literature review and reasoning why Unet was selected. There is so much better autoencoder like models Unet++ and many more and none were mentioned ir the literature review. The literature review must be improved.
* In detail provides the mathematical formulation about the six channels, halftones and their representation.
* Figure 2 what are the 0,1,2,3 ? It should be explained in the caption.
* Error in Eq (2) y_i^i -> y_i^p, even better notion for predictions \hat{y}_i 
* Please remove Table 1. Everyone knows what the basic unet is.
* Table 2 - 2 numbers after the comma is sufficient
* In all comparisons authors near Yik's method should add the baseline model of just image decomposition to halftones palette colors by weighting palette color via linear combination.

Reviewer 4 Report

Dear Authors,

The proposed research is fascinating and practical. However, I am afraid to let you know that I could not find the novelty of this research. Most importantly, the literature review is insufficient. Many researchers have used a modified version of U-Net architecture for image watermarking purposes. What makes your proposed model different is not highlighted in your manuscript.  I am strongly suggesting that you should re-consider doing more literature reviews and perform more comparisons to highlight the novelty of your proposed research. Good luck!

Regards,

Reviewer

Round 2

Reviewer 2 Report

The paper was improved and now can be accepted.

Reviewer 3 Report

charactersl -> characteristics

framework[22, 26, 27] -> framework [22, 26, 27]

"An example result of using U-Net to quantize to 0-1 and then
77 expanded to 0-3 is shown in Fig 1c." -> delete, there should be no references to text in introduction.

Fig 2a, Fig 2b, Fig 2c, Fig 2d should be cited and explained in text.

method[X] -> method [X]

F-norm should be provided in manuscript.

Reviewer 4 Report

Dear Author,

I believe the manuscript has been sufficiently improved to warrant publication in Electronics. Good work! 

Regards, 

Reviewer
